# Dual Role of p73 in Cancer Microenvironment and DNA Damage Response

**DOI:** 10.3390/cells10123516

**Published:** 2021-12-13

**Authors:** Julian M. Rozenberg, Svetlana Zvereva, Alexandra Dalina, Igor Blatov, Ilya Zubarev, Daniil Luppov, Alexander Bessmertnyi, Alexander Romanishin, Lamak Alsoulaiman, Vadim Kumeiko, Alexander Kagansky, Gerry Melino, Nikolai A. Barlev

**Affiliations:** 1Laboratory of Cell Signaling Regulation, Moscow Institute of Physics and Technology, 141701 Dolgoprudny, Russia; sv.zvereva.2014@gmail.com (S.Z.); iblatov@bk.ru (I.B.); ilyamitozubarev@gmail.com (I.Z.); luppov.dv@phystech.edu (D.L.); lamak.alsuliman@gmail.com (L.A.); kagasha@yahoo.com (A.K.); 2The Center for Precision Genome Editing and Genetic Technologies for Biomedicine, Engelhardt Institute of Molecular Biology, Russian Academy of Sciences, 119991 Moscow, Russia; alexandra.dalina@gmail.com; 3Faculty of Computer Science, School of Software Engineering, Higher School of Economics University, 101000 Moscow, Russia; abessmertnyi@hse.ru; 4School of Life Sciences, Immanuel Kant Baltic Federal University, 236041 Kaliningrad, Russia; romanishin.alexander97@yandex.ru; 5School of Biomedicine, Far Eastern Federal University, 690091 Vladivostok, Russia; vkumeiko@yandex.ru; 6Department of Medicine, University of Rome Tor Vergata, 00133 Roma, Italy; Gerry.melino@uniroma2.it; 7Institute of Cytology, Russian Academy of Science, 194064 Saint-Petersburg, Russia; 8Institute of Biomedical Chemistry, 119435 Moscow, Russia

**Keywords:** immuno-oncology, transcription, p53, p73, tumour microenvironment, epithelial-mesenchymal transition, EMT

## Abstract

Understanding the mechanisms that regulate cancer progression is pivotal for the development of new therapies. Although p53 is mutated in half of human cancers, its family member p73 is not. At the same time, isoforms of p73 are often overexpressed in cancers and p73 can overtake many p53 functions to kill abnormal cells. According to the latest studies, while p73 represses epithelial–mesenchymal transition and metastasis, it can also promote tumour growth by modulating crosstalk between cancer and immune cells in the tumor microenvironment, M2 macrophage polarisation, Th2 T-cell differentiation, and angiogenesis. Thus, p73 likely plays a dual role as a tumor suppressor by regulating apoptosis in response to genotoxic stress or as an oncoprotein by promoting the immunosuppressive environment and immune cell differentiation.

## 1. Introduction

Alterations of p73 activities are associated with cancer hallmarks including cell cycle regulation, replicative immortality, and genomic instability. However, p73-driven changes of these properties are not sufficient for cancer formation per se and can be detected in normal cells as well. The other factors crucial for tumour development is tumour escape from the recognition by immune cells and the formation of the pro-inflammatory tumour microenvironment. It is worth noting that the ability of tumour cells to invade surrounding tissues and metastasise is the main cause of death from cancers. 

As extensively described, the tumor suppressor p53, often dubbed as “genome guardian”, is involved in the regulation of these hallmarks of cancer. p53 belongs to the family of transcription factors that comprises three proteins: p53, p63, and p73 [1,2,3]. p63 is the master regulator of epidermal development and homeostasis [4,5,6], a process largely dependent on complex enzymatic processing [2,7,8,9,10,11,12,13]. All three members of the p53 family are present in tumour cells and are expressed as multiple isoforms, which often play opposite functions in tumorigenesis.

While p53 is lost or mutated in most tumours, mutations in the Tp73 gene are relatively rare [14,15,16,17,18,19]. On the other hand, specific p73 isoforms are overexpressed in a variety of malignancies and serve as biomarkers of aggressiveness of the disease [20,21,22,23,24,25,26,27]. 

Based on the domain structure, the p73 isoforms can be roughly divided into two classes (Figure 1). Transcriptionally competent p73 isoforms with the full-length N-terminal domain are transcribed from the first external promoter of the Tp73 gene [28,29,30]. The second intragenic promoter of the Tp73 gene produces so-called DN-isoforms (DNp73) lacking the transcriptional activator domain (TAD) but retaining their DNA binding domains. Since the DNA binding domain is conserved within the p53 family, p73 can bind the promoters of the subset of the p53 target genes regulating essential mechanisms in the cell cycle (p21), apoptosis (Bax, Fas, PUMA), and metabolism [31,32,33,34,35,36]. In general, DN-isoforms repress transcription of the p53/TAp73 target genes [37,38,39], but in combination with other proteins, they can activate a set of genes that is very different from those activated by the TAp73 [40,41,42,43,44]. Both TAp73 and p53 activate the DNp73 promoter creating a negative feedback loop [45,46,47].

Therefore, transcriptionally active p73 isoforms (TAp73) generally possess pro-apoptotic and anti-oncogenic properties, while the N-terminally truncated isoforms (DNp73) tend to be anti-apoptotic and pro-oncogenic [24,48,49,50,51,52,53,54,55].

At the C- terminus of p73, alternative splicing generates a variety of isoforms: α, β, γ, δ, ε, and ζ [26,56] (Figure 1). The full-length p73α isoform contains a distally located sterile alpha motif (SAM) domain that is absent in p53. This domain allows p73 to act both as a transcriptional repressor by preventing interaction with p300/CBP [57] and as a transcriptional co-activator by promoting interactions with c-Jun, Nfkb, and ATF3 transcription factors [58,59,60,61,62,63,64].

Despite certain structural differences, the p73 and p53 proteins have a high degree of homology. Moreover, the signaling pathways that regulate p73 and p53 are also similar [65,66]. For example, the DNA damage response pathway activates both proteins [67]. Furthermore, similar to p53, the p73 transcription activity is repressed by the MDM2 E3 ligase [68,69,70,71]. However, in contrast to p53, this interaction does not result in p73 degradation [72,73,74,75]. Therefore, the loss of p53 functions in cancer can be partially compensated by TAp73 [76,77,78,79].

As a result of their structural similarity, certain p53 mutants interact with p73, causing dysregulation of p73 and hence, affecting the antitumor effect [80,81]. Furthermore, the wild-type p53 has been recently shown to interact with p73 as well, inducing apoptosis via JNK-induced p53 Thr81 phosphorylation [82]. Similarly, TAp73a and the Δ133 isoform of p53 act synergistically to promote the expression of several DNA repair genes (RAD51, LIG4, and RAD52) through binding to the p53-responsive elements in the promoters of these genes [83]. In turn, the DNp73 isoform represses the ATM-p53 mediated DNA damage response directly at the sites of DNA damage [84]. Overexpression of DNp73 in tumours can inhibit the DNA binding and tumour-suppressive functions of p53 leading to cancer progression [48,50,85]. Thus, the precise functional outcome of the p73-p53 interaction depends on which particular isoform of each protein participates in the interaction, thus providing yet another layer of complexity to the regulation of the p73/p53 network in cancer.

The later stages of cancer are associated with well-known cancer hallmarks [86] including induction of angiogenesis to feed the growth of cancer cells [87,88], activation of metastasis driven by the epithelial-to-mesenchymal transition (EMT) [89,90,91], evasion from the immune surveillance, and formation of the tumor microenvironment [92,93,94,95,96,97]. 

In this review, we summarised the literature data suggesting that while repressing epithelial-mesenchymal transition and metastasis in some cancers, the p73 transcriptional activity can promote cancer growth by modulating the cytokine excretion of cancer and immune cells, thereby promoting polarisation of immune cells to tumorigenic phenotypes in tumour microenvironment and inducing angiogenesis. Thus, we speculate that p73 is likely to play a dual role in tumorigenesis acting as a tumour suppressor by regulating apoptosis in response to genotoxic stress and as an oncoprotein promoting the immunosuppressive environment and immune cell differentiation.

## 2. Angiogenesis Induction

Angiogenesis is the process of forming new blood vessels from pre-existing vessels. This process is vital for tumour development since cancer cells actively proliferate and require ample nutrition and oxygen supply [98]. Several studies assessed the role of p73 in angiogenesis (Figure 2).

One of the central angiogenic regulators is the transcriptional factor HIF-1α (hypoxia-inducible factor 1α). Under normoxic conditions, HIF-1a is rapidly degraded due to post-translational modifications and stabilizes upon hypoxia. Importantly, hypoxia is the condition that is prevalent in solid tumors. TAp73 has been demonstrated to inhibit angiogenesis via HIF-1α degradation by an E3 ubiquitin ligase, MDM2 [99]. Another study showed that the loss of TAp73 and overexpression of the DNp73 isoform resulted in elevated angiogenic activity, i.e., augmented HIF-1α-mediated expression of proangiogenic cytokines [88]. These data were obtained using the mice and zebrafish models as well as analysis of publicly available breast cancer RNA sequencing data. These two studies suggest that TAp73 can reduce the HIF-1α activity to attenuate the angiogenic phenotype, which was prevented by DN-p73.

On the other hand, p73 itself is stabilised by different mechanisms upon hypoxia. Experiments on several cancer cell cultures of different origins with different statuses of p53 and p73 revealed that under hypoxic conditions, the TAp73 was stabilised by HIF-1α, which suppressed the E3 ligase Siah1. Siah1 was shown to mediate the p73 degradation [100]. The ChIP assay on MEFs demonstrated that TAp73 bound to the promoter of the proangiogenic VEGF-A (Vascular endothelial growth factor A) gene and activated its expression. In contrast, another study found that under normoxic conditions, p73 inhibited VEGF expression. Importantly, these results were obtained using two p53-deficient cell lines (Hek-293 and Saos2) [101], which raises the question of whether the p53 status plays any role in the outcome of p73 action.

In line with this, the loss of or mutations in the p53 gene induced the antiangiogenic activity of p73. On the contrary, the presence of wild-type p53 leads to the proangiogenic activity of p73 in HCT116 cells [102]. The p73 overexpression increases VEGF production, concomitantly decreasing the expression of one of the major negative regulators of the angiogenesis thrombospondin-1 (TSP-1). TSP-1 repression was observed in p73 overexpressing ovarian carcinoma cells A2780 bearing wild-type p53. Furthermore, p73 overexpressing tumours showed elevated vascularisation. These data are consistent with recent research, which revealed a positive correlation between the expression of TAp73, mutant p53, and vasohibin-1. Vasohibin-1 is an antiangiogenic factor that inhibits the VEGF expression in lung adenocarcinoma [103]. Summarising these studies, it is tempting to speculate that p53 status affects the role of TAp73 in angiogenesis in a compensatory way, but this hypothesis requires additional research.

Besides HIF-1α, an adenosine monophosphate-dependent protein kinase (AMPK), which is activated by hypoxia, also stabilises TAp73 [104]. In hypoxia, blunting the AMPK signaling has been shown to reduce the protein level of TAp73 by allowing Siah1 to ubiquitinylate p73. In contrast, activation of AMPK alone in the absence of hypoxia maintains the TAp73 protein without repressing Siah1. However, the AMPK-mediated stabilisation of TAp73 without hypoxia may be insufficient to induce the tube formation by endothelial cells [104].

Like TAp73, DNp73 can be stabilised under hypoxic conditions [105]. DNp73 was found to be preserved by HIF-1α, a master regulator of transcription under hypoxic conditions. The latter acts upstream of Siah1 to abolish its destabilising activity against DNp73 in hypoxia. HIF-1α regulates the expression of vascular endothelial growth factor, VEGF, which, in turn, controls angiogenesis (Figure 1). Since p73 was implicated in vasculogenesis, it was decided to investigate the regulation of VEGF-A expression by DNp73 [106]. To this end, the researchers studied mouse embryonic fibroblasts lacking the DNp73 protein. The endogenous expression of VEGF-A in these cells was reduced both at the basal level and under hypoxic conditions. After transfection with various p73 plasmids, analysis of endogenous VEGF-A showed that DNp73β was able to induce VEGF-A expression at the level of RNA promoting tumour vascularisation. In glioblastoma, where DNp73 is overexpressed, DNp73 regulates the expression of ANGPT1/Tie2, enhancing angiogenesis and tumour growth, which, in turn, can be inhibited by the treatment with the selective inhibitor of the TIE2 kinase rebastinib [53].

Another study looked at the role of p73 in the formation of the vascular plexus and angiogenesis [106,107]. This paper demonstrated both in vivo and in vitro that DNp73 affected the migration of endothelial cells and tubulogenesis, regulating the expression of VEGF and TGFβ [106]. Moreover, enhanced tumor vascularisation was observed in xenografts of B16-F10 melanoma cells overexpressing DNp73.

An inspection of 112 colon tumour clinical samples and corresponding normal tissues also showed the positive correlation between p73 and VEGF expression [108]. Noteworthy, the expression of VEGF165b, which is considered to perform the antiangiogenic function, was also found to be upregulated by different p73 isoforms. This study also revealed the correlation between ΔEx2p73 and PEDF (Pigment Epithelium-Derived Factor), which plays an antiangiogenic role in many cancer types. Another clinical data analysis showed the correlation between the p73 and VEGF expression as well as the tumour vascular density in colorectal carcinoma [109].

Therefore, the role of p73 in cancer angiogenesis is an important direction of further research. Previous studies have shown that p73 can be stabilised by HIF-1α and AMPK, and that both DNp73 and TAp73 upregulate the expression of proangiogenic genes. Yet, at the same time, TAp73 can induce HIF-1α degradation inhibiting angiogenesis (Figure 2). The angiogenic activity of p73 probably depends on the p53 status. The dominant role of p73 in the HIF-1α-mediated induction of angiogenesis has been suggested to explain an intricate relationship between p73 and angiogenesis [110].

## 3. Activation of Invasion and Metastasis

Cancer metastasis is a multistep process of invasion, intravasation, dissemination, and extravasation of cancer cells from the primary tumour site followed by the growth at a new location [111]. Metastasis is the leading cause of death among cancer patients. Considering that expression of the pro-oncogenic p73 isoforms often correlates with overall patient survival [55,112,113,114,115], it is possible that p73 also plays a role in metastasis. One of the first steps associated with cancer invasion is the epithelial-to-mesenchymal transition (EMT). During this process, the proteins responsible for tight cell contacts in epithelial tissue are downregulated (E-Cadherin) and substituted for mesenchymal markers, including N-Cadherin, vimentin, and SM-actin. At the same time, the levels of several metal-dependent proteases are augmented to mediate remodelling of the extracellular matrix (ECM). This step is crucial for escaping tumour cells from the site of a primary tumour and invading the adjacent tissues. The key regulators of EMT genes are transcription factors Slug, Snail, Twist, and Zeb1. Surprisingly, the role of p73 in this process is poorly investigated and likely depends on the cancer type (Figure 3). 

In the three-dimensional model of the MCF10A breast cell line, p73 knockdown led to EMT, which was exemplified by downregulation of E-cadherin and concomitant upregulation of EMT transcription factors [116]. In addition, TAp73 depletion resulted in the disruption of normal cell polarity in spheroids, which is critical for the initiation of EMT [116,117,118]. Moreover, using three-dimensional cultures of Madin-Darby canine kidney (MDCK) cells, TAp73, p21, and PUMA were shown to play a critical role in cyst formation while TAp73 knockdown led to EMT by decreasing E-cadherin and increasing Snail and Twist [119].

Consistent with the repressive role of TAp73 in EMT, TAp73 depletion was shown to increase cell migration of colorectal cancer HCT116 and lung carcinoma H1299 cells [120]. Mechanistically, downregulation of the HDAC1-HSP90 signalling promoted the proteasomal degradation of TAp73 [120]. Among the p73 targets is the NAV3 gene, which encodes a microtubule-binding protein that plays a crucial role in the p73-mediated inhibition of cell migration [89]. The EMT markers and metalloproteinases MMP2 and MM9 were also induced by depletion of either NAV3 or p73 [89].

In pancreatic cancer, the gain-of-function mutant p53 induces the PDGFRb expression causing cell invasion in vitro and formation of metastasis in vivo. This happens likely via dissociation of the p73/NF-Y complex [121]. Conversely, the PDGFRb expression is inhibited in the p53-deficient non-invasive cells. Thus, TAp73a inhibits invasion of the pancreatic cancer cells via the protein-protein interaction with NF-Y [121]. Similarly, the TAp73 loss in the murine model of pancreatic ductal adenocarcinoma was associated with EMT induction and shorter survival [122]. Remarkably, EMT induction was mediated by depletion of the secreted protein biglycan (BGN), a TGFb inhibitor. In turn, BGN was induced by TGFb, suggesting a negative feedback loop between them [123]. Moreover, TGFb promoted Treg function and immunosuppression in pancreatic cancer [124]. 

In the FaDu pharynx squamous cell carcinoma cells and the corresponding xenograft model, the Abrus agglutinin (AGG) treatment induced p73 and suppressed Snail and EMT [125]. Intriguingly, p73 was shown to interact with Snail upon EGF treatment, while AGG inhibited this interaction. This caused cytoplasmic translocation of Snail and its subsequent proteasomal degradation repressing EGF-induced EMT [125]. 

Ras-dependent activation of phosphatidylinositol-3 kinase (PI3K) signalling was shown to promote a switch from the TA- to DN-p73 isoforms facilitating the anchorage-independent growth of transformed fibroblasts. In line with this, the ectopic expression of either H-RasV12WT, or H-RasV12Y40C repressed TAp73 via the PI3K activation in the HCT116 colon cancer cells [126].

Recently, in melanoma, DNp73 was shown to downregulate EPLIN (Epithelial protein lost in neoplasm), leading to the activation of the IGF1R-AKT/STAT3 pathway, depletion of E-cadherin, induction of Slug, invasion and metastasis [127]. 

The same group reported a correlation between high levels of DNp73 and aggressiveness of melanoma with hypopigmentation [128]. Mechanistically, DNp73 activates the IGF1R/PI3K/AKT pathway leading to proteasomal degradation of tyrosinase (a key enzyme in the melanin synthesis), induction of Slug, and EMT. These EMT events correlated with higher invasion allowing for stratifying the patients according to their hypopigmentation status [128].

The p73 protein is known to regulate the genes responsible for metastasis, but the direct association is not always established. For example, HMGB1, a putative p73 target that can regulate p73-mediated transactivation [129,130], is associated with EMT, metastasis, and poor prognosis in breast, lung, and other cancer types [55,131,132,133]. 

In contrast to the role of p73 as a repressor of EMT, in several cancer cell lines p73 was shown to promote expression of the Integrin-β4 gene, which protein product enhances EMT [134]. Further, overexpression of the direct p73 target, POSTN, that encodes the integrin-binding protein restores the invasion ability of glioblastoma cells after p73 depletion [135]. 

Therefore, the role of p73 in the regulation of EMT and invasion is cancer type-specific. While p73 inhibits EMT and invasion of pancreatic [121,122], colon [89], breast, lung [116,120], and oral carcinoma cells [125], it can also promote invasion of glioblastoma cell lines [135] (Figure 3). In addition, DNp73 induces invasion of hypopigmented melanoma cells [127,128].

## 4. Tumour-Promoting Inflammation and Evading Immune Destruction

Tumour microenvironment is determined by crosstalk between different signalling pathways taking place in cancer cells, immune cells, and other cancer-associated cells such as fibroblasts [96,136,137,138,139,140,141,142,143].

Evidence for the role of p63/p73 in the immune regulation by “non-immune“ cells first came from the study of the autoimmune disease palmoplantar pustulosis. The elevated levels of p63 and p73 proteins were shown to contribute to the enhanced IL-6 production by the reticular crypt epithelial cells that surround lymphoid follicles [144]. The IL-6 mediated activation of the JAK/STAT3 pathway is pro-tumorigenic and also facilitates Th17 T cell differentiation. The latter, in combination with TGFb signaling, can be pro- or anti-carcinogenic by promoting the immune response via the recruitment of CD8 T cells [145]. Both p63 and p73 were demonstrated to activate the IL-6 promoter using overexpression experiments and reporter plasmids. However, functional experiments using p73 depletion were not performed in this study [144].

In light of the role of p73 in the regulation of the immune response, it should be noted that in glioblastomas, p73 regulates POSTN expression [135] which in turn promotes M2 macrophage polarisation and tumour growth [146,147].

The inflammasome is a protein complex that functions as a sensor of pathogen infections [148]. The aberrant inflammasome activation can promote skin and breast cancer but repress colon cancer [149]. The inflammasome signals to the innate immune cells by caspase-1-mediated cleavage and secretion of the IL-1β. Exogenous expression of TAp73b has been recently shown to upregulate the IL-1β expression both at the mRNA and protein levels in H1299 cells [150]. 

Another report described the role of p73 in atopic dermatitis, a long-term skin inflammation [151]. The DNp73/TAp73 ratio was reported to be increased in the hyperplastic keratinocytes. Paradoxically, this led to enhanced production of the thymic stromal lymphopoietin protein (TSLP), a cytokine that causes Th2 cells-mediated allergic inflammation and also induces the antigen presentation by Langerhans cells [152,153]. In addition, TSLP activation was mediated by exogenous expression of DNp73, concomitantly with an increase of NF-kB, a known p63/p73 coregulator [151,154,155] (Figure 4A). Activation of TSLP signalling promoted the Th2 immune response in breast [92] and pancreatic [156] cancers and was associated with poor prognosis [157]. 

In breast cancer, TSLP induced the anti-apoptotic factors Bcl-2 and Bcl-xL [130] as well as several p73 transcriptional targets related to inflammation, including VEGF, TSLP, IL-1β, and IL-6. These genes were downregulated by the Bcl-2 inhibitor ABT-737 in the activated human mast cell line [158]. Therefore, it is possible that p73 plays a regulatory role in production of these cytokines in immune cells. Further investigations are required to validate this hypothesis. 

Another cytokine controlled by p73 is the tumor necrosis factor-alpha (TNFa), which is produced by macrophages [159]. p73 repressed TNFa-induced apoptosis of the immune cells [160,161,162], thereby promoting the immunosuppressive cancer microenvironment. The TNFa-induced apoptosis is mediated via degradation of Rb, thereby leading to activation of the c-Abl/p73 cascade. This mechanism is executed in various cancer cell lines including colorectal carcinoma cells, mice fibroblasts, and thymocytes [163]. In contrast, in the p53 null squamous cell carcinoma cells, TNF-α promoted c-REL nuclear translocation, dissociation of TAp73 from DNp63α, and nuclear export of TAp73 to the cytoplasm [61]. As a result, TNF-α modulates a genome-wide redistribution of the DNp63α/TAp73 and DNp63α/c-REL complexes from the TP53 to AP-1 DNA binding sites to induce an oncogenic gene expression program in squamous cancer cells [63]. 

As mentioned above, accumulating evidence suggests that TNFa typically promotes apoptosis in normal cells by activating p73. For example, p73 promotes apoptosis induced by the TNFa/IL-1 in the KRT15+ epithelial stem cells [164]. In turn, specific p73 isoforms can also regulate TNFa-induced apoptosis of endothelial cells [165]. The authors demonstrated that TNFa specifically induced the pro-apoptotic TAp73a but at the same time repressed the anti-apoptotic DNp73 form [165]. The p73 −/− mice were used to evaluate the effect of p73 on the TNFa-induced apoptosis [163,164]. p73 −/− mice exhibit a dysregulated immune system and develop chronic inflammation in the epithelial organs, including rhinitis, otitis, periorbital edema, and conjunctivitis [166]. The p73 −/− mice also demonstrated the disrupted development of retinal vasculature. Experiments on cell cultures showed that the absence of p73 affected the production of angiogenic factors to promote the tumour suppressive microenvironment [88,106,141,167,168]. Thus, p73 isoforms regulate apoptosis and differentiation of the innate and adaptive immune cells thereby determining the tumour microenvironment [159,169,170,171,172]. 

TAp73 depletion in breast cancer cells was associated with their more aggressive growth when xenografted into mice [97]. Mechanistically, this effect was attributed to the TAp73-dependent repression of secretion of chemoattractant Ccl2. This event led to the recruitment of CD204+ and CD206+ pro-carcinogenic macrophages [97]. 

Collectively, p73 likely affects the tumour microenvironment by regulating the expression of cytokines. However, to unravel the exact effect of p73 on the tumour microenvironment, a comprehensive cytokine profiling of cells with different statuses of p73 is required.

The major factor responsible for tumour promoting inflammation is macrophage polarisation [173,174]. In the course of studying the inflammatory response after lipopolysaccharide (LPS) challenge, it was found that the lack of TAp73 in macrophages prolonged the M1-type macrophage polarisation. These results were obtained using the TAp73 knockout mice and wild-type mice, where endogenous macrophages were substituted by the TAp73-/- ones [159]. In vitro, TAp73-/- macrophages produced more pro-inflammatory cytokines such as TNFa, IL-6 and macrophage inflammatory protein-2. In addition, TAp73-/- macrophages displayed a decreased phagocytosis and increased major histocompatibility complex class II expression. Thus, the lack of p73 prolonged the survival of pro-inflammatory macrophages [159]. These results predict that in tumour-associated macrophages, TAp73 will likely promote M2 polarisation and immunosuppressive tumour microenvironment (Figure 4B).

In adaptive immunity, CD4+ Th1 cells can reprogram M2-like peritoneal exudate cells into the M1 phenotype upon the MHCII-specific interaction. This event enhances an accumulation of M1-polarised tumour-associated macrophages [170]. Th1/Th2 polarization contributes to the immunosuppression environment and progression in gliomas [175,176], hepatocellular carcinoma [177] melanoma [178], and breast cancer [179,180]. Both TA- and DN-isoforms of p73 have been recently shown to negatively affect Th1 differentiation of the naive splenic CD4 T-cells and repress the expression of the IFNg gene [171]. Thereby, it is reasonable to speculate that in the tumour microenvironment, p73 expression in the CD4 T-cells suppresses Th1 polarisation and promotes Th2 polarisation, M2-like macrophages, and immunosuppressive tumour microenvironment.

The same stimuli that activate p53 are likely to activate p73 as well [181]. If true, then p73 should induce the T-cell death and attenuate their well-documented antitumour effects in nearly all cancer types [182,183,184]. Noteworthy, TNFa-induced T-cell apoptosis was dependent on the p73, but not p53 [163]. Importantly, in the cycling peripheral T cells, apoptosis induced by binding of the T-cell receptor (TCR) was independent of p53 but depended on TAp73 [163]. Upon TCR activation, T cell survival was regulated by the MDM2-mediated p73 repression [185]. Inhibiting the MDM2-p73 interaction activated the p73-driven expression of the proapoptotic gene Bim and subsequent apoptosis [185]. Therefore, developing therapeutic strategies that would selectively activate p53 but not p73, would help induce cell death in tumour cells, sparing the T cells from elimination and contributing to the overall anticancer immune response [186].

## 5. Conclusions

The p53 protein exerts a crucial inhibitory role in cancer development [200,201,202,203,204]. p73 is involved in tissue morphogenesis [205,206] and is rarely mutated in tumours; however, specific isoforms are overexpressed in lung, brain, breast, and other cancer types [207,208]. Here, we reviewed the influence of p73 on the relations between cancer cells and the immune system from the point of view of cancer hallmarks. While the DNA damage response and genomic stability, ability to sustain proliferation, and metabolic control are extensively studied, relatively little is known about the impact of p73 on the ability of cancers to metastasise and the role of p73 in the forming immunosuppressive tumour microenvironment (Table 1). 

An analysis of the most recent research reveals that while repressing the epithelial-mesenchymal transition and metastasis in some cancers, the p73 transcriptional activity can promote cancer growth via several mechanisms. For example, p73 directly enhances the Warburg effect stimulating glycolysis in cancer cells. Furthermore, p73 modulates the secretomes of cancer and immune cells to promote the tumour microenvironment. The latter subsequently affects polarisation of immune cells towards the tumorigenic phenotype, culminating in the induction of angiogenesis. Thus, p73 plays a dual role in acting as a tumour suppressor by regulating apoptosis in response to genotoxic stress and as a pro-oncogene by promoting immunosuppressive environment and immune cell differentiation. The experimental evidence reviewed in this study favours the hypothesis that at the advanced stages of cancer development, tumour cells may employ p73 to repress the immune system surveillance explaining why p73 is rarely mutated. This possibility should be taken into account when developing new anti-cancer therapeutic strategies.

## Figures and Tables

**Figure 1 cells-10-03516-f001:**
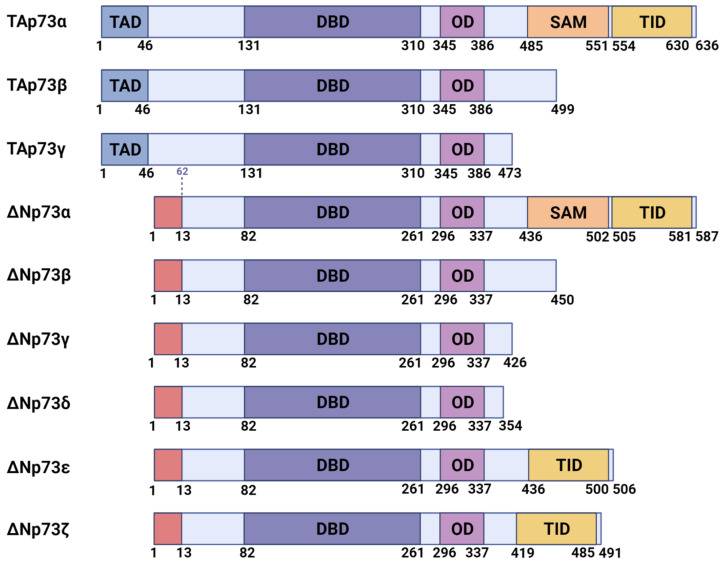
Major isoforms of p73 and their domain structure. TAD, transactivation domain, present in the transcriptionally active (TA) isoforms with a full-length N-terminus. N-terminally truncated (DN) isoforms have a distinct N-terminus (the first 13 N-terminal amino acids). DBD, DNA-binding domain; OD, oligomerization domain; SAM, sterile alpha motif; TID, transcriptional inhibitory domain.

**Figure 2 cells-10-03516-f002:**
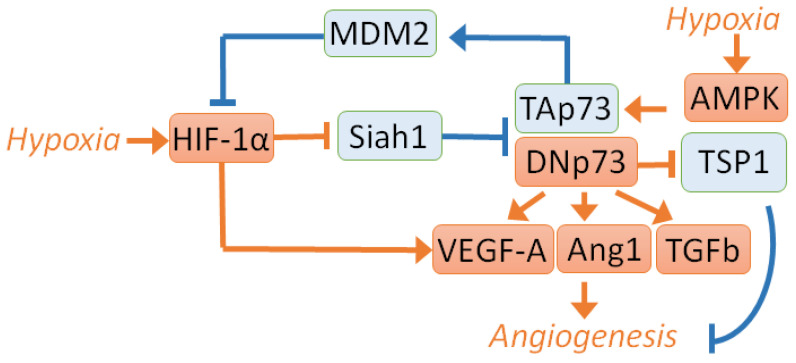
Regulation of angiogenesis by p73. The antiangiogenic signaling is shown in blue, and proangiogenic signalling is shown in red. TAp73 inhibits angiogenesis via MDM2-mediated degradation of HIF-1α [99], whereas the loss of TAp73 and overexpression of DNp73 results in the increased HIF-1α activity and upregulation of proangiogenic cytokines [88]. In turn, TA- and DN- p73 isoforms are stabilised by HIF-1α-dependent suppression of the Siah1-mediated degradation during hypoxia [100]. Similarly, TAp73 can be stabilised by AMPK [104]. Consequently, DNp73 or TAp73 can target VEGF-A, Ang1, and TGFb to promote angiogenesis. On the contrary, TSP-1 is repressed by DNp73, which leads to enhanced angiogenesis in xenograft cancer models [53,102,106].

**Figure 3 cells-10-03516-f003:**
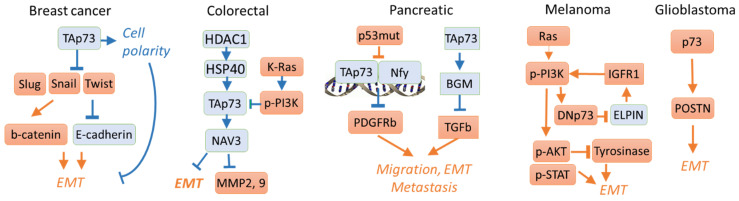
The p73-regulated signalling pathways influence EMT, mobility, or invasiveness in cancer models. In blue shown is the EMT repressive and in red—the EMT-promoting signalling. TAp73 inhibits EMT, mobility, and invasion for breast [116], colorectal [89,120,126] or pancreatic cancers [121,122]. However, the induction of DNp73 inhibits EMT and metastasis in melanoma [127,128]. In glioblastoma, total p73 depletion induces an invasive phenotype in the POSTN-dependent manner [135].

**Figure 4 cells-10-03516-f004:**
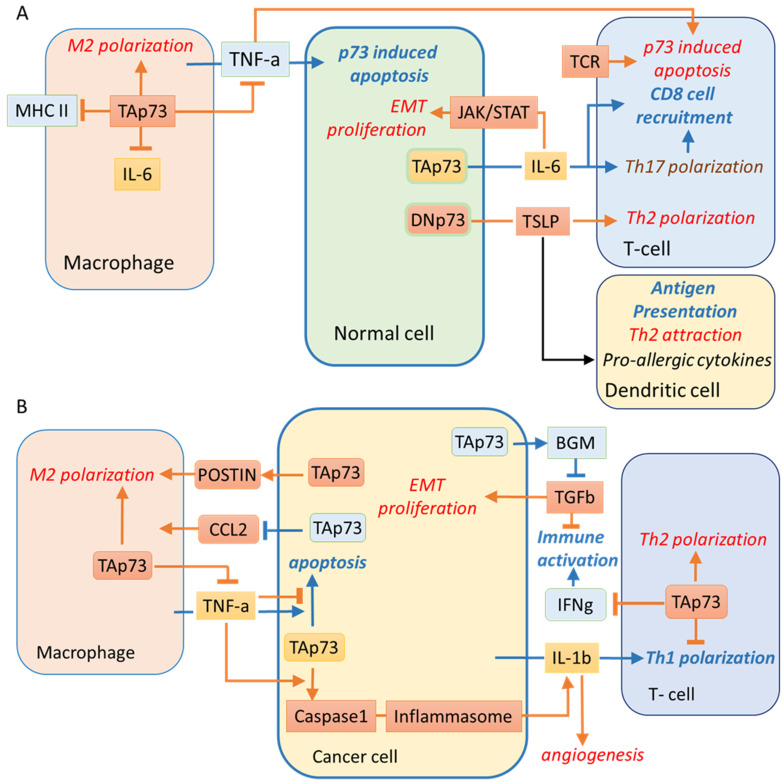
The p73-regulated signalling in normal, cancer, and immune cells. TAp73 in cancer cells and immune cells promotes the immunosuppressive cancer environment. In blue shown is cancer-repressing and in red—cancer-promoting signalling, respectively. Arrows represent the positive association, and pointers with blocks are negative associations with green and red anti- and pro-cancerous signalling, respectively. Yellow boxes represent dual signalling molecules with pro- and anticancerous properties. (**A**) In the normal cells, p73 promotes the IL6 production [144], which can have either pro- [187,188], or anti- cancerogenic effects by inducing Th17 v.s. Treg polarization [189,190]. In skin tissues, p73 regulates TSLR [151] that induces Th2 cell differentiation, thereby promoting chronic inflammation in atopic dermatitis and is associated with poor prognosis in several cancers [157]. (**B**) In breast cancer, TAp73 represses CCL2 secretion and pro-cancerogenic macrophage recruitment [97]. In contrast, TSLR induces p73 targets VEGF, TSLR, IL-1b, and IL-6 hence promoting breast cancer angiogenesis. In turn, IL1b has a plethora of pro-cancerogenic effects [191]; however, it can also induce the anti-cancerogenic Th1 polarisation [192] and is regulated by the p73/caspase1/inflammasome axis [150,193]. p73-mediated caspase-1 expression is induced by TNFa that also induces a pro-cancerogenic program in SSC [63] but apoptosis in endothelial cells, respectively [165,194,195,196]. The TAp73-induced expression of BGN inhibits the TGFb pro-oncogenic signalling [122]. In contrast, the p73-mediated Warburg effect [197] promotes acidosis, enhanced invasiveness [198], and M2 macrophage polarisation [93]. In turn, the p73 signalling in macrophages promotes M2 polarisation and represses TNF-a production, also promoting phagocytosis and lowering the MHC2 expression [159]. Noticeably, TNFa- or strong TCR-binding-induced apoptosis in T-cells was dependent on p73, but not p53 [163,199]. Moreover, in T-cells, TAp73 and DNp73 repress Th1 polarisation and IFNg production, promoting the immunosuppressive environment [171].

**Table 1 cells-10-03516-t001:** The cancer hallmarks and p73 functions in specific cancers. The table structure: The rightmost column represents the Tp73 expression in tumour tissue (Tp73Tum) in comparison with the normal tissue (Tp73Norm) as a ratio: log2(Tp73Tum + 1)/log2(Tp73Norm + 1) [209].

Cancer	Invasion and Metastasis	Angiogenesis	Inflammation	Role of p73 Isoforms in Cancer	Relative tp73 Expression in Cancer Tissue
Non-small cell lung carcinoma (NSCLC)	[134]		[150]	[210]	[211]
Lung adenocarcinomas		[103]	[193]	[99,212]	2, 95
Hepatocellular carcinoma (HCC)					
Cervix carcinoma	[126]			[213,214,215]	15, 2
Melanoma	[126,128]				
Osteosarcoma		[99,100,101]			5, 4
Glioblastoma	[126,135]		[135]	[216]	5, 2
Medulloblastoma				[217]	---
B-cell lymphoma		[88]		[218]	33, 6
Breast cancer	[116]	[88,100]	[150]	[219]	1, 2
Colorectal cancer	[89]	[88,100,108,109]	[163]	[55,220]	9, 6
Esophageal adenocarcinoma	[125]			[221]	1, 1
Thyroid cancer					
Ovarian cancer		[102]		[222]	4, 1
Pancreatic cancer	[121]		[156]	[223,224]	3, 15
Neuroblastoma	[135]				
Squamous carcinoma			[61,63]	[225,226]	1, 26

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
