# Peer review of "Dual Role of p73 in Cancer Microenvironment and DNA Damage Response"

_cells, 2021, doi:10.3390/cells10123516_

Round 1
Reviewer 1 Report
This is a very nicely written review about p73 in the tumor microenvironment and immune regulation. I believe the topic is timely and suitable for publication in the journal. It deserves to be published as soon as possible so that the field can benefit from this review.
Author Response
Thank you very much.
Sincerely.
Julian Rozenberg, PhD
Reviewer 2 Report
This manuscript reviews an interesting and topic, concerning the involvement of p73 on cancer progression, through the manipulation of the tumor microenvironment and also on immune suppression, in processes not necessarily related to cancer. There are no recent reviews regarding these subjects. However, a series of concerns and recommendations should be addressed for the manuscript to be accepted for publishing:
First, the title should be less evasive, more informative, presenting the general concept observed by the authors.
Please proofread the text for grammar and syntax errors, including commas, the use of the article "the", typos and En dashes and hyphens on phrasal objects. Please pay attention to words that were crossed out and not removed from the text upon submission of the manuscript.
This reviewer feels the necessity of further explanations in the introduction, for a more comprehensible exposure of the subject. Authors should provide general information about the themes mentioned in the title (tumor microenvironment, its key players, for instance). Also, it is important to introduce readers to the roles of the p53 family members in cancer.
p53 relationship with p73 is mentioned very superficially in the manuscript, such as in page 3, line 111. A paragraph in the introduction section should be included to clarify this interplay for readers. It is important to mention that there is an interaction between p73 and mutant p53, but not with the p53 wild type form. For this reason, there are critical differences in the scenarios in which p53 is WT, mutated and also when it is deleted. Please consider these different backgrounds in the first description of the subject and in the statements made throughout the text.
Classic literature describing the Hallmarks of Cancer by Hanahan and Weinberg are not cited in this manuscript, although it is used as the backbone to group the evidences.
Since they are mentioned throughout the text, it is very important that authors include a scheme describing the p73 isoforms mentioned, for a better understanding by readers.
page 2 line 73: authors should include here that p73 is only able to interact with p53 in the mutant form, through their DNA-binding sites, which could be related to the effects observed in a mut-p53 background.
page 2 line 92: is this affirmation related to solid tumors?
page 2 line 98: since mice were called mice, please change "D. rerior" to "zebrafish", to reach a larger number of readers, or vice-versa.
page 3, line 117: correct "53".
Author Response
Reviewer 2
We would like to thank the reviewer for careful reading of our manuscript, constructive comments, and suggestions.
We re-wrote the introduction describing the interaction between p53 and p73. According to the reviewer’s suggestion, we incorporated an additional figure with the domain structures of p73 isoforms.
We also included the description of angiogenesis, metastasis, and cancer microenvironment. However, due to extensive editing of the whole text, the exact changes were not marked.
Please find below point-by-point answers to the reviewer’s comments:
First, the title should be less evasive, more informative, presenting the general concept observed by the authors.
According to the reviewer’s suggestion we have changed the title to: “Dual role of p73 in cancer microenvironment and DNA damage response.”
Please proofread the text for grammar and syntax errors, including commas, the use of the article "the", typos and En dashes and hyphens on phrasal objects. Please pay attention to words that were crossed out and not removed from the text upon submission of the manuscript.
We have carefully checked the text for grammatic mistakes and other typos.
This reviewer feels the necessity of further explanations in the introduction, for a more comprehensible exposure of the subject. Authors should provide general information about the themes mentioned in the title (tumor microenvironment, its key players, for instance). Also, it is important to introduce readers to the roles of the p53 family members in cancer.
We appreciate the reviewer for this suggestion. We have included several paragraphs into the introduction section, which provide the information on angiogenesis, metastasis, and cancer microenvironment supported by the relevant references (page 2 lines 6-17, page 3 lines 8-14, page 4, lanes 1-4).
p53 relationship with p73 is mentioned very superficially in the manuscript, such as in page 3, line 111. A paragraph in the introduction section should be included to clarify this interplay for readers. It is important to mention that there is an interaction between p73 and mutant p53, but not with the p53 wild type form. For this reason, there are critical differences in the scenarios in which p53 is WT, mutated and also when it is deleted. Please consider these different backgrounds in the first description of the subject and in the statements made throughout the text.
We agree with the reviewer and have now included relevant paragraphs in the introduction section describing the interactions between p73 and various forms of p53 and their different functional outcomes (Highlighted with yellow in the text page 3, lines 15-27).
Classic literature describing the Hallmarks of Cancer by Hanahan and Weinberg are not cited in this manuscript, although it is used as the backbone to group the evidences.
We apologize for this mistake and have now cited this paper in the introduction section.
Since they are mentioned throughout the text, it is very important that authors include a scheme describing the p73 isoforms mentioned, for a better understanding by readers.
We included such a scheme shown in Figure 1 of the revised version of the manuscript.
page 2 line 73: authors should include here that p73 is only able to interact with p53 in the mutant form, through their DNA-binding sites, which could be related to the effects observed in a mut-p53 background.
Done (Page 2, line73)
page 2 line 92: is this affirmation related to solid tumors?
This is correct.
page 2 line 98: since mice were called mice, please change "D. rerior" to "zebrafish", to reach a larger number of readers, or vice-versa.
We have changed this abbreviation into the more common one.
page 3, line 117: correct "53".
Corrected.
Round 2
Reviewer 2 Report
Alterations accepted